# Clinically Isolated β-Lactam-Resistant Gram-Negative Bacilli in a Philippine Tertiary Care Hospital Harbor Multi-Class β-Lactamase Genes

**DOI:** 10.3390/pathogens12081019

**Published:** 2023-08-08

**Authors:** Alecks Megxel S. Abordo, Mark B. Carascal, Roland Remenyi, Doralyn S. Dalisay, Jonel P. Saludes

**Affiliations:** 1Clinical and Translational Research Institute, The Medical City, Pasig 1605, Philippines or alecksabordo@gmail.com (A.M.S.A.); mbcarascal@themedicalcity.com (M.B.C.); rgremenyi@themedicalcity.com (R.R.); 2Center for Chemical Biology and Biotechnology (C2B2) and Department of Biology, University of San Agustin, Iloilo 5000, Philippines; ddalisay@usa.edu.ph; 3Balik Scientist Program, Philippine Council for Health Research and Development, Department of Science and Technology, Taguig 1631, Philippines; 4Center for Natural Drug Discovery and Development (CND3) and Department of Chemistry, University of San Agustin, Iloilo 5000, Philippines

**Keywords:** β-lactamase, antibiotic resistance, Gram-negative bacilli, Philippines

## Abstract

In the Philippines, data are scarce on the co-occurrence of multiple β-lactamases (BLs) in clinically isolated Gram-negative bacilli. To investigate this phenomenon, we characterized BLs from various β-lactam-resistant *Klebsiella pneumoniae*, *Escherichia coli*, *Acinetobacter baumannii*, and *Pseudomonas aeruginosa* isolated from a Philippine tertiary care hospital. The selected Gram-negative bacilli (*n* = 29) were resistant to either third-generation cephalosporins (resistance category 1 (RC1)), cephalosporins and penicillin-β-lactamase inhibitors (RC2), or carbapenems (RC3). Isolates resistant to other classes of antibiotics but susceptible to early-generation β-lactams were also selected (RC4). All isolates underwent antibiotic susceptibility testing, disk-diffusion-based BL detection assays, and PCR with sequence analysis of extended-spectrum BLs (ESBLs), metallo-BLs, AmpC BLs, and oxacillinases. Among the study isolates, 26/29 harbored multi-class BLs. All RC1 isolates produced ESBLs, with *bla*_CTX-M_ as the dominant (19/29) gene. RC2 isolates produced ESBLs, four of which harbored *bla*_TEM_ plus *bla*_OXA-1_ or other ESBL genes. RC3 isolates carried *bla*_NDM_ and *bla*_IMP_*_,_* particularly in three of the metallo-BL producers. RC4 *Enterobacteriaceae* carried *bla*_CTX-M_, *bla*_TEM_, and *bla*_OXA-24-like_, while *A. baumannii* and *P. aeruginosa* in this category carried either *bla*_IMP_ or *bla*_OXA-24_. Genotypic profiling, in complement with phenotypic characterization, revealed multi-class BLs and cryptic metallo-BLs among β-lactam-resistant Gram-negative bacilli.

## 1. Introduction

Among the top-priority multidrug-resistant (MDR) pathogens identified by the World Health Organization (WHO), β-lactam- (i.e., carbapenem-) resistant Gram-negative bacilli, specifically *Acinetobacter baumannii*, *Pseudomonas aeruginosa*, and members of the *Enterobacteriaceae* family, are the most critical [1]. In Southeast Asia, the emergence of these critical pathogens has been reported in the last ten years [2]. Resistance to β-lactam antibiotics or their combination with β-lactamase inhibitors (i.e., clavulanic acid, sulbactam) has been steadily increasing in low- to middle-income countries (LMICs) like the Philippines. In particular, the Department of Health’s (DOH) antimicrobial resistance surveillance program (ARSP) noted up to 50% carbapenem resistance rates among Gram-negative bacilli from Philippine hospitals in 2022. Similarly, among *Enterobacteriaceae*, the ARSP noted up to 47% and 78% resistance to cephalosporins and penicillins, respectively [3]. The WHO predicted that the deaths associated with MDR pathogens could reach ten million by 2050 [4]. Comprehensive investigations of the resistant pathogens are necessary for determining potential ways to combat the AMR onslaught.

Ongoing AMR surveillance programs exist globally (Global Antimicrobial Resistance and Use Surveillance System (GLASS, WHO) [5]) and in the Philippines (ARSP, DOH) [3]. However, information on the prevailing mechanisms of resistance among MDR pathogens is lacking. Extensive molecular genotyping studies on MDR Gram-negative pathogens in the Asia–Pacific were conducted in the past ten years, but representation from the Philippines is low [6]. Previous Philippine studies reported that the production of extended-spectrum β-lactamases (ESBLs) and carbapenemases (CPs) remains a widespread resistance mechanism among clinically isolated Gram-negative bacilli [3,7]. As such, the local health department recommended genotyping and real-time clinical data analysis as measures for the timely isolation of patients and infection control among hospitals with ESBL- and CP-producing bacteria [3]. The reports on the occurrence of β-lactamases (BLs) are limited by the scarcity of molecular characterization studies among Philippine isolates. As β-lactams alone and in combination with BL inhibitors remain to be in the empiric antibiotic recommendations for Gram-negative pathogens, the significantly increasing trend of resistance against these antibiotics presents challenges for clinicians, microbiologists, and epidemiologists. Specifically, uncharacterized β-lactam resistance could lead to incorrect therapeutic decisions by infectious disease specialists, discrepant laboratory testing results (laboratory phenotype vs. actual isolate phenotype in the patients), and difficulty in monitoring the spread of resistance genes [8].

BLs are enzymes capable of hydrolyzing the chemical bonds of β-lactam antibiotics. Currently, there are thousands of discovered BLs, both chromosomal and acquired [9]. Acquired BLs present a significant challenge since genes related to them can be easily transferred on intra- and inter-species levels which complicates infection control [8]. The global BL database (BLDB) categorized BLs into classes based on their DNA sequence identities and protein structures [9]. The Ambler classification system defines class A as serine penicillinases and ESBLs, class B as metallo-BLs mostly hydrolyzing carbapenems, class C as AmpC cephalosporinases, and class D as oxacillinases [10]. The BLs produced by the pathogens dictate the choice of antibiotics against the infections or provide information on the spread of outbreaks [11,12]. Therefore, the characterization of these BLs is vital in creating appropriate guidelines for MDR treatment and control.

Contemporary strains of MDR Gram-negative bacilli harbor multiple types of BLs, which present challenges in the clinical management of patients [13]. In particular, the expression of multiple BLs is implicated in false antibiotic susceptibility profiles [14] and the masking of other BLs (i.e., ESBL detection masked by other metallo-BLs) [15]. Clinical laboratories in the Philippines rely on phenotypic assays to detect these BLs. However, these assays have inherent limitations in their diagnostic sensitivity and do not distinguish the different BL classes [16]. Genotypic testing allows for the accurate identification of BL genes [17]. In the Philippines, clinically isolated Gram-negative bacilli were genotyped previously, but the findings were limited to a few selected BL genes, including Ambler class A *bla*_CTX-M_*, bla*_TEM_*, bla*_SHV_ [7,18], class B *bla*_NDM_, *bla*_VIM_ [19], and class D *bla*_OXA-1_ [20]. Therefore, herein we report the characterization of multiple classes of BLs among clinically isolated β-lactam-resistant Gram-negative bacilli in a Philippine tertiary care hospital. Specifically, we highlighted the importance of genotyping in characterizing MDR pathogens in the Philippines, complementing the phenotypic assays currently used in the hospital setting.

## 2. Materials and Methods

### 2.1. Collection and Characterization of Bacterial Isolates

Gram-negative bacteria, identified using Vitek^®^ MS (bioMérieux, Marcy-l’Étoile, France) as *E. coli*, *K. pneumoniae, P. aeruginosa*, and *A. baumannii*, were collected at The Medical City (TMC), a private tertiary care hospital (500-bed capacity) located in National Capital Region of the Philippines. The isolates came from diverse clinical samples, including endotracheal aspirates, sputum, urine, blood, and wounds. The selected isolates were chosen based on their antimicrobial susceptibility testing (AST) profile from Vitek^®^ 2 Compact (bioMerieux, France) following the latest Clinical and Laboratory Standards Institute (CLSI) breakpoints [21]. Specifically, the pathogens matched the resistance categories (RC) set using clinically relevant criteria deduced from the latest ARSP report [3] and described in the literature [22,23,24,25]. RC1 refers to the isolates which were non-susceptible to third-generation cephalosporins. This resistance phenotype represents about 38–45% of *E. coli* and *K. pneumoniae,* 15% of *P. aeruginosa*, and 51% of *A. baumannii* local isolates [3]. RC2 refers to those non-susceptible to third-generation cephalosporins and penicillin-β-lactamase inhibitors, which represent about 25% of *E. coli*, 36% of *K. pneumoniae*, 15% of *P. aeruginosa*, and 50% of *A. baumannii* local isolates [3]. RC3 are those resistant to carbapenems and other β-lactams. This phenotype represents up to 9% of *E. coli*, 16% of *K. pneumoniae*, 15% of *P. aeruginosa*, and 52% of *A. baumannii* local isolates [3]. Finally, RC4 refers to isolates with unusual resistance profiles (e.g., levofloxacin-resistant but ertapenem-susceptible *Enterobacteriaceae* [24]; carbapenem-resistant but cephalosporin-susceptible *P. aeruginosa* and *A. baumannii* [25]). These unusual yet important phenotypes are not captured by the ARSP report and may be missed by most local hospitals. For clarity, we defined non-susceptible isolates as those with intermediate susceptibility or resistance in the AST based on the CLSI breakpoints [21]. The number of study isolates was limited to the first two bacteria per resistance category and per species, prospectively collected between January and October 2021. A total of twenty-nine isolates (two *E. coli* and *K. pneumoniae* per category, two *A. baumannii* each for RC1-RC3, while only one in RC4, and two *P. aeruginosa* each for RC2-RC4) matched the resistance categories and were used for the study. All isolates were maintained and grown in Tryptic Soy agar (Merck-Millipore, Darmstadt, Germany) and incubated at 35 °C.

### 2.2. Detection of β-Lactamase Production

The study isolates were profiled for ESBL and CP production based on the high-confidence phenotypes in Vitek^®^ 2 Advanced Expert System (AES, bioMérieux, Marcy-l’Étoile, France). Standard phenotypic assays were also performed to confirm the production of ESBLs and CPs among the isolates [21]. These assays served as orthogonal methods to the phenotypes generated by the Vitek^®^ 2 AES.

#### 2.2.1. Disk-Diffusion-Based ESBL Test

Individual isolates were standardized to 0.5 McFarland using a SPECTROstar Nano^®^ UV–Vis spectrophotometer (BMG Labtech, Ortenberg, Germany). The standardized isolates were then streaked on Mueller–Hinton agar (MHA; Merck-Millipore, Darmstadt, Germany), and antibiotic disks were placed using the standard disk diffusion procedures. The antibiotic disks contained ceftazidime, cefotaxime (30 µg each; Mastdiscs AST, Liverpool, UK), ceftazidime–clavulanate, and cefotaxime–clavulanate (30 µg/10 µg each combination; HiMedia, Mumbai, India). After 16–18 h of incubation at 35 °C, an increase of at least 5 mm in zones of inhibition (ZOIs) between the standalone antibiotic disk and the disk with antibiotic plus clavulanate indicated ESBL production. *K. pneumoniae* ATCC^®^ 700603 and *E. coli* ATCC^®^ 25922 were used as controls.

#### 2.2.2. Modified Carbapenem Inactivation (mCIM), and Ethylenediamine Tetraacetic Acid (EDTA)-Modified Carbapenem Inactivation Methods (eCIM)

The individual isolates (1–10 µL loopful) were emulsified in two sets of 2 mL Tryptic Soy broth (Merck-Millipore, Darmstadt, Germany; one with EDTA for eCIM, and one without for mCIM) with meropenem disks (10 µg; Mastdiscs AST, Liverpool, UK). After four hours of incubation at 35 °C, the disks were transferred to MHA plates with standardized *E. coli* ATCC^®^ 25922. After 16–18 h of incubation at 35 °C, a 6–15 mm ZOI in mCIM setups indicated positive serine CP production. Isolates with a positive mCIM assay result and a ≥5 mm increase in ZOI between mCIM and eCIM setups were classified as metallo-BL producers. *K. pneumoniae* ATCC^®^ BAA-1705 (serine CP producer), ATCC^®^ BAA-2146 (metallo-BL producer), and ATCC^®^ BAA-1706 (non-CP producer) were used as controls.

### 2.3. Molecular Characterization of Study Isolates

The genomic DNA of all selected isolates was extracted using the PureLink™ Genomic DNA Mini kit (Invitrogen, Carlsbad, CA, USA). GeneExplorer^®^ thermocycler (Bioer, Zhejiang, China) was used for all polymerase chain reaction (PCR) assays. PCR targeting the 16S rRNA and various BL genes (Ambler class A *bla*_CTX-M_, *bla*_KPC_, *bla*_SHV_, *bla*_TEM_; class B *bla*_IMP_, *bla*_NDM_, *bla*_VIM_; class C *bla*_DHA_, *bla*_FOX_; and class D *bla*_OXA-1_, *bla*_OXA-2_, *bla*_OXA-23-like_, *bla*_OXA-24-like_, *bla*_OXA-48-like_, *bla*_OXA-51-like_, and *bla*_OXA-58-like_) was performed for all isolates. The target BL genes were limited to the commonly acquired genes as reported in previous studies [26,27,28]. All PCR reactions were conducted in singleplex using a 25 µL reaction mix comprising of 12.5 µL GoTaq^®^ Green Master Mix (Promega, Madison, WI, USA), 0.5–1.0 µL 10 μM forward and reverse primers, 50–100 ng DNA template, and nuclease-free water. The primer sequences, positive controls, expected product sizes, and reference protocols for the PCR assays were listed in Appendix A [29,30,31,32,33,34,35,36,37,38]. Amplicons were visualized in 1.5–2% (*w/v*) agarose gels (1st BASE, Singapore) run on a MyGel Instaview^®^ electrophoresis system (Accuris, Ansan, South Korea) with Tris–acetate–EDTA buffer (Vivantis, Selangor, Malaysia) at 135 V for 30 min. Distinct amplicons (PCR products, excised gels) with expected product sizes were sent for bidirectional Sanger sequencing using the forward and reverse primers described (Macrogen, Seoul, Republic of Korea). To confirm negative PCR results, we tested all isolates with the Xpert^®^ Carba-R PCR test in GeneXpert^®^ IV System (Cepheid, Sunnyvale, CA, USA) following the manufacturer’s protocol. The test specifically detects the presence of Amber class A *bla*_KPC_, class B *bla*_NDM_*, bla*_VIM_*, bla*_IMP_, and class D *bla*_OXA-48-like_.

### 2.4. Sequence Analysis of the 16S rRNA and β-Lactamase Genes

Consensus DNA sequences were generated from the high-quality forward and reverse sequences of each detected gene using Bioedit version 7.2.5. All DNA sequences were initially analyzed using the Basic Local Alignment Search Tool (nucleotide BLAST) for the 16S rRNA sequences, and β-lactamase database BLAST (http://bldb.eu/ accessed on 17 January 2023) for BL gene sequences. All sequences with confirmed identity were curated at the GenBank^®^ database and assigned with accession numbers (Appendix A). To determine the relationship of the gene sequences to their closest references, we constructed neighbor-joining and maximum-likelihood trees. ClustalW, jModelTest version 20160303, PAUP* version 4b10, and PhyML version 3.1 were used for multiple sequence alignment, optimal model testing, neighbor-joining tree, and maximum likelihood tree construction, respectively. TreeExplorer version 2.12 was used for phylogenetic tree visualization. The reference sequences and parameters used for the tree construction were listed in Appendix A. Phylogenies were used to infer the closest potential identities of the genes based on clustering with known reference genes.

## 3. Results

### 3.1. Selected Gram-Negative Bacilli Exhibit Varying Resistance to β-Lactams, and Produce ESBLs and CPs

The 29 isolates had confirmed species identities based on their 16S rRNA gene sequences (Appendix A) and exhibited clinically relevant resistance profiles against different β-lactams (Figure 1 and Appendix A). All the chosen isolates conformed with the chosen resistance categories (RC1–RC4) for the study. Although the Vitek^®^ 2 profiles of the isolates indicated resistance to β-lactams, the production of BLs potentially responsible for the resistance phenotypes was not directly detected in AST. Vitek^®^ 2 AES provided high-confidence phenotypes for all the isolates. The results showed the production of ESBLs by the isolates from all resistance categories, while CP production was limited to all RC3 isolates, and RC4 *A. baumannii* and *P. aeruginosa*. To corroborate these results, standard phenotypic assays on ESBL and CP production were performed in all isolates (Figure 1, Appendix A). The assays confirmed ESBL production in 22/29 isolates, with negative phenotypic assay results occurring mostly in *A. baumannii*. Only five of eight putative CP producers were confirmed in the phenotypic assay, with the variance existing only in *A. baumannii* and *P. aeruginosa* isolates. The phenotypic assays also detected the production of metallo-BLs, with four out of eight RC3 isolates (*E. coli* and *K. pneumoniae*) testing positive for the enzyme.

### 3.2. β-Lactam-Resistant Gram-Negative Bacilli Harbor Multi-Class BL Genes

To determine the type of BLs produced by the isolates, we tested the entire panel for the presence of common BL genes using PCR (Figure 2, Appendix A) and inferred gene identities using phylogenetic sequence analysis (Figure 2, Appendix A). Among the Ambler class A BL genes tested in this study, only *bla*_CTX-M_, *bla*_TEM_, and *bla*_SHV_ were confirmed to be present. The *bla*_CTX-M_ genes were detected in 19/29 isolates, with *bla*_CTX-M-117_ as the most common related variant (detected in 14 isolates). All *E. coli* and *K. pneumoniae* study isolates, regardless of their resistance phenotype, harbored the *bla*_CTX-M_ gene. The *bla*_TEM_ genes were also detected in 18/29 isolates, with *bla*_TEM-20_ as the common related variant (detected in 11 isolates). Similar to *bla*_CTX-M_, *bla*_TEM_ was also detected in almost all *E. coli* and *K. pneumoniae* study isolates (except one *K. pneumoniae* under RC1). Meanwhile, *bla*_SHV_ genes were only detected in 13/29 isolates, mainly in *K. pneumoniae* and *A. baumannii*. The *bla*_SHV-5_ was common in *A. baumannii*, while *bla*_SHV-27_, *bla*_SHV-1_, and *bla*_SHV-5_ were detected in *K. pneumoniae*. In terms of the Ambler class B genes tested, only *bla*_NDM_ and *bla*_IMP_ were confirmed to be present. The *bla*_NDM_ genes were detected in 19/29 isolates, distributed in all the species regardless of their resistance phenotype. The *bla*_NDM-1_ gene was the most common related variant (detected in 14 isolates). The *bla*_IMP_ genes were detected in 14/29 isolates from all selected species. The *bla*_IMP-1-like_ gene was the most common related variant (detected in eight isolates). A total of 24/29 test isolates harbored *bla*_NDM_ or *bla*_IMP_ genes including some carbapenem-susceptible isolates.

Among the Ambler class C genes tested, only *bla*_DHA_ genes were confirmed to be present. The *bla*_DHA-23_-related variant was only detected in two *E. coli* isolates (one each from RC1 and RC2). In terms of the Ambler class D genes tested, only *bla*_OXA-1_, *bla*_OXA-24-like_, and *bla*_OXA-51-like_ were detected among the study isolates. The *bla*_OXA-1_ genes were confirmed in 19/29 isolates, comprising eight *E. coli*, seven *K. pneumoniae*, and four *P. aeruginosa* regardless of their resistance phenotypes. Meanwhile, *bla*_OXA-24-like_ genes were confirmed in 18/29 of the isolates, distributed regardless of the resistance phenotype. The *bla*_OXA-72_ was the related variant detected in all the isolates. Finally, *bla*_OXA-51-like_ genes were only detected in all seven *A. baumannii* study isolates, with *bla*_OXA-66_ and *bla*_OXA-68_ as the common related variants. The complementary Xpert^®^ Carba-R PCR test to detect common CP genes (Amber class A *bla*_KPC_, class B *bla*_NDM_, *bla*_VIM_, *bla*_IMP_, and class D *bla*_OXA-48-like_) confirmed some results of our PCR and sequence analyses (Appendix A). For instance, none of the study isolates harbored the *bla*_VIM_, *bla*_KPC_, and *bla*_OXA-48-like_ genes after both singleplex PCR with sequencing and CARBA-R tests. Meanwhile, the CARBA-R PCR test could not detect *bla*_IMP_ in any isolate, and *bla*_NDM_ in only three study isolates (in contrast to fourteen and nineteen detected in PCR with sequencing, respectively, for each gene). Other tested genes, such as *bla*_FOX_, *bla*_OXA-2_, *bla*_OXA-23-like_, and *bla*_OXA-58-like_, were not detected in any of the isolates even after repeated PCR using appropriate controls.

Among the test isolates, 26/29 harbored BL genes from at least two BL classes. Notably, one *E. coli* each from RC1 and RC2 carried BLs from all classes despite being resistant only to cephalosporins and penicillin-β-lactamase inhibitor combinations based on AST. A total of 15 out of 16 phenotypically assessed ESBL-producing *E. coli* and *K. pneumoniae* study isolates also harbored Ambler class B or D BLs. Similarly, 10 of 11 CP-producing study isolates based on phenotype carried Ambler class A in combination with class B or D BL genes. Only three of twenty-four study isolates with class B BL genes tested positive for the metallo-BL phenotypic test. *E. coli* and *K. pneumoniae* from RC4 harbored Ambler class B and D CP genes despite being susceptible to carbapenems based on AST. Meanwhile, one *P. aeruginosa* and *A. baumannii* each from RC4 carried Ambler class A ESBL genes despite being susceptible to cephalosporins.

Comparing the distributions of the BL genes revealed patterns when grouped based on their resistance category, bacterial species, and BL production (Figure 3). For example, while most of the BL genes tested in this study were shared in all resistance categories, *bla*_DHA_ was only detected in one of the RC1 and one RC2 *E. coli* isolates. In terms of the distribution of the BL genes per bacterial species, *bla*_TEM_ was only detected in *E. coli*, *K. pneumoniae*, and *P. aeruginosa* study isolates. In contrast, *bla*_SHV_ was only detected in *K. pneumoniae*, *P. aeruginosa*, and *A. baumannii* study isolates. *bla*_OXA-1_ was only found in *E. coli*, *K. pneumoniae*, and *P. aeruginosa*, and *bla*_OXA-51-like_ only in *A. baumannii* study isolates. Lastly, the *bla*_DHA_ was only reported in ESBL-producing study isolates, while the rest of the detected genes were found in both ESBL and CP producers.

## 4. Discussion

The analysis of the burden of AMR in 204 countries reported β-lactam resistance as the highest contributor to mortality from all leading MDR pathogens [39]. Generally, β-lactam resistance involves complex mechanisms, including efflux overexpression, porin loss or modification, and production of degradative enzymes [8]. Among these mechanisms, the production of β-lactam-degrading BLs remains the most significant in Gram-negative bacilli [40]. This fact is supported by the recent ARSP report which indicated that up to 47% of multidrug-resistant *Enterobacteriaceae* (*K. pneumoniae*, *E. coli*) are ESBL producers based on routine phenotypic laboratory tests [3]. Our results reported the BL profiles of selected β-lactam-resistant Gram-negative bacilli isolated from a tertiary hospital in the Philippines. In particular, the co-harboring of multiple BLs among the resistant isolates was highlighted. The co-existence of these enzymes will continue to rise among Gram-negative bacilli due to the widespread exchange of mobile genetic elements between species [40].

Prior to discussing our detailed findings, it is important to note that our study isolates are not necessarily exact representations of all the antibiotic-resistant strains in the Philippines. Rather, our isolates represent the generally reported, clinically important resistance phenotypes among Gram-negative isolates as recently reported in ARSP [3]. In addition, RC4 isolates (unusual resistance) were included to capture rare phenotypes which may be missed in routine clinical laboratory analyses and in the ARSP report. We acknowledge that strain-to-strain variabilities in genotypes exist within isolates of the same phenotype. Hence, our results could only provide preliminary insights into the uninvestigated characteristics of the local strains, which could serve as foundations for further in-depth investigations in the locality. The readers must take caution in interpreting our data for epidemiological purposes.

In terms of phenotypically characterizing the isolates, our results showed discrepancies in ESBL and CP production when comparing the Vitek^®^ 2 AES-suggested phenotype versus the result of the disk-diffusion-based assays. This non-concordance has already been reported previously in assessing ESBL production in *Enterobacteriaceae* [41] and CP production among MDR pathogens [42], with the standard phenotypic assays having higher accuracy. The variance is attributed to the overprediction of the Vitek 2^®^ AES algorithm. The discrepancy may also be due to the low specificity of the disk-diffusion-based assays when used in certain species. For instance, CLSI only recommends the phenotypic ESBL test in *Enterobacteriaceae* [21]. Currently, there is no gold-standard method for ESBL detection in *A. baumannii* and *P. aeruginosa*. Similarly, mCIM and eCIM were only recommended for *Enterobacteriaceae* and *P. aeruginosa*, not for *A. baumannii* [21]. Our results showed that phenotypic test discrepancies were only observed in *A. baumannii* (negative in phenotypic ESBL test, mCIM, and eCIM) and *P. aeruginosa* (negative in mCIM and eCIM), corroborating the CLSI recommendations. In addition, other mechanisms of carbapenem resistance not measured by the chosen phenotypic tests, including high levels of chromosomal BLs expression, porin loss, or efflux pumps [43], may have contributed to the discrepancy between the assay results and the isolate’s resistance profiles.

Non-susceptibility to third-generation cephalosporins (i.e., ceftazidime, ceftriaxone) is mostly attributed to the production of ESBLs and AmpC cephalosporinases. The RC1 isolates used in our study captured this resistance phenotype. All study isolates in this category were ESBL-producers, with *bla*_CTX-M_ as the common Ambler class A gene. This result confirms the conclusions of previous studies on the prevalent ESBLs among Philip-pine clinical isolates [18,19]. Most isolates under this resistance category harbored multiple BL classes, including class B metallo-BLs in *E. coli* and *A. baumannii,* class C AmpC in *E. coli*, and class D oxacillinases in all species. Despite the presence of CP-encoding genes, these isolates were not resistant to carbapenems nor phenotypically producing CPs. This result indicated that certain CP genes in the β-lactam-resistant study isolates were possibly cryptic. In the environment, bacteria carrying cryptic BL genes serve as silent spreaders of antibiotic resistance [44]. In the clinical setting, metallo-BL genes in meropenem-susceptible Gram-negative bacilli were already reported [35]. Cryptic antibiotic resistance genes are hypothesized to be expressed only in prolonged exposure to their target antibiotics [44]. This phenomenon presents a challenge in the therapeutic management of patients. Specifically, pathogens harboring cryptic metallo-BLs may be carbapenem-susceptible in standard AST but turn out to be resistant after antibiotic therapy [45]. Therefore, caution must be observed when reporting clinical AST results that only consider phenotypic data.

Some cephalosporin-resistant isolates may also exhibit non-susceptibility to penicillin-β-lactamase inhibitors (i.e., amoxicillin–clavulanate, piperacillin–tazobactam), hence the inclusion of RC2 in our study. Resistance to both cephalosporins and combination antibiotics may be induced by class A *bla*_TEM_, class C AmpC, and some class D oxacillinases [46]. Our results showed that almost all the isolates in RC2 possessed *bla*_TEM_ (usually co-occurring with other ESBLs and oxacillinases) which supports the established resistance mechanism. Penicillin-β-lactamase-inhibitor-resistant isolates co-harboring *bla*_OXA-1_, *bla*_CTX-M,_ and *bla*_TEM_ appeared to be also common. *bla*_OXA-1_ (encoding for inhibitor-resistant β-lactamase) has been reported to co-exist with other ESBL genes (*bla*_TEM_, *bla*_CTX-M_), potentially reducing the activity of clavulanate in combination therapies [47]. Like the observations in RC1 isolates, metallo-BLs were also detected in most RC2 isolates despite being carbapenem-susceptible.

The latest ARSP reported increasing rates of carbapenem-resistant Gram-negative bacilli in the Philippines [3]. RC3 represents these MDR pathogens. These isolates are already resistant to cephalosporins and penicillin-β-lactamase inhibitors, so they are expected to co-harbor multiple BLs [46]. Production of CPs is the major mechanism of carbapenem resistance, and all our RC3 isolates produce the enzyme (either as serine- or metallo-BLs). The most prevalent types of CPs are the acquired ones, which are mostly the gene targets in our genotyping assays. Among the CP genes detected, *bla*_NDM_, *bla*_IMP_, and *bla*_OXA-24-like_ were common, although not exclusive in carbapenem-resistant pathogens. Only the RC3 *E. coli* and *K. pneumoniae* study isolates phenotypically produced metallo-BLs. Upon examining the class B genes detected in these isolates, all possessed *bla*_NDM_, some co-existing with *bla*_IMP_. The *bla*_NDM_ has been increasingly reported among carbapenem-resistant *E. coli* and *K. pneumoniae* isolates in the Philippines [19,48,49]. Our finding that *bla*_NDM_ may also be cryptic in carbapenem-susceptible isolates could potentially explain the spread of the gene in the clinical setting.

Unusual resistance phenotypes involving β-lactams are also observed in the hospital setting. Among *Enterobacteriaceae*, a fluoroquinolone-resistant but ertapenem-susceptible phenotype is considered unusual [50]. In the literature, AmpC BLs (*bla*_CMY_, *bla*_MIR_) were described in these isolates [50]. Our results indicated that multi-class BLs (*bla*_TEM_, *bla*_SHV_, *bla*_OXA-1_) were present in these isolates. These genes may be attributed to the cephalosporin resistance of the isolates (in addition to the unusual phenotype). As with the other non-carbapenem-resistant study isolates, these pathogens also harbored the cryptic metallo-BLs. We did not detect any of the tested class C AmpC BLs in these isolates, suggesting that other BL or non-BL mechanisms might be responsible for the resistance profile. Meanwhile, among *A. baumannii* and *P. aeruginosa*, a carbapenem-resistant but cephalosporin-susceptible phenotype is unusual. Typically, resistance to carbapenems is also associated with resistance to other β-lactams (penicillins, cephalosporins, and penicillin-β-lactam-inhibitors) given the wide spectrum of β-lactam substrates degradable by most CPs [39]. Hence, the emergence of carbapenem-resistant strains still susceptible to cephalosporins presents a challenge to clinicians in terms of choosing the appropriate antibiotic for therapy [25]. Reporting of this phenotype is very sparse, and the resistance mechanism is usually attributed to efflux overexpression or mutations in the *oprD* (encoding for outer membrane protein) [51,52]. This phenotype was only reported in the literature for *P. aeruginosa* [51,52]. We reported carbapenem-resistant ceftazidime-susceptible *A. baumannii* for the first time in the Philippines. Unlike the previous reports of non-involvement of CPs for this resistance phenotype, the category D *A. baumannii* and *P. aeruginosa* study isolates carried *bla*_IMP_ or *bla*_OXA-24-like_ genes. Further investigation into other related mechanisms for this unusual resistance profile (i.e., porin-mediated resistance) should also be considered.

Overall, our study has been limited to the number and types of isolates characterized, the resistance categories identified, the choices of phenotypic tests, and the target genes analyzed. In addition, our study did not investigate non-acquired-BL-related mechanisms of resistance (i.e., overexpression of chromosomal BLs and efflux pumps, porin loss and modification), nor assess the whole genome characteristics of the isolates which could have explained the discrepant results between the phenotypes and genotypes. Despite these limitations, the extensive characterization methods employed could still provide important information on the resistance phenotypes and genotypes of the selected isolates. Our results produced significant findings, which included: the detection of multi-class BLs among β-lactam-resistant Gram-negative bacilli in a Philippine hospital, discovering the presence of cryptic metallo-BLs among certain carbapenem-susceptible isolates, and the initial report of unusual carbapenem-resistant but cephalosporin-susceptible *A. baumannii*.

## 5. Conclusions

Our findings provided insights into the importance of genotyping in characterizing MDR pathogens in the Philippines, complementing the phenotypic assays currently used in the hospital setting. We highlighted the co-harboring of multiple BL genes which may induce resistance phenotypes with implications for infection management and control. The study also uncovered cryptic metallo-BLs among carbapenem-susceptible study isolates, which presents a challenge in the current AST interpretation in the hospital setting. Extensive profiling of clinically relevant pathogens and determining the occurrence of multiple BLs is important since it may help predict treatment responses to various β-lactam antibiotics. Given the limitations of the current phenotypic tests in determining the presence of ESBLs and CPs in Gram-negative bacilli, genotyping remains a reliable predictor of the resistance phenotype. Therefore, future studies may focus on expanding the target genes tested, potentially revealing new insights into the dynamics of BL-induced β-lactam-resistance among Gram-negative bacilli. Investigating the interconnected antibiotic resistance pathway (not just β-lactam-resistance but also resistance to other antibiotics) through whole genome analysis may also help in painting a bigger picture of their underlying resistance mechanisms. Finally, increasing the number of representative isolates per clinically relevant resistance category may help further confirm the significant findings of this study.

## Figures and Tables

**Figure 1 pathogens-12-01019-f001:**
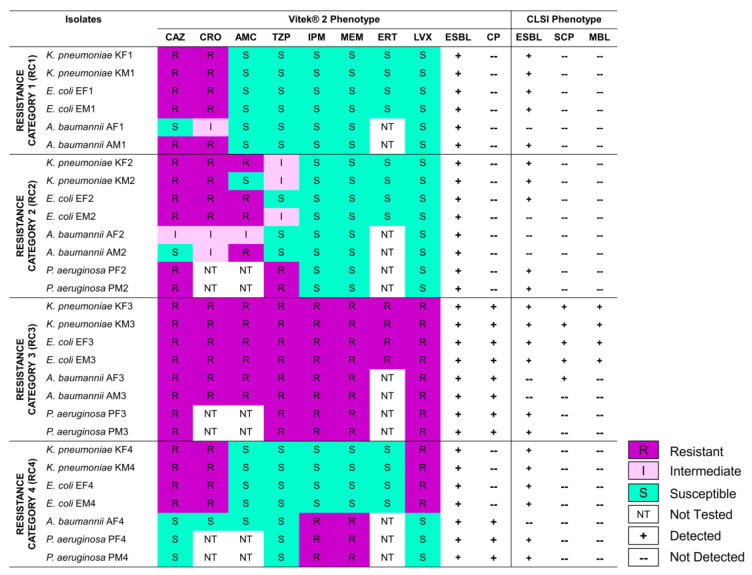
Summarized antibiogram results from Vitek^®^ 2 Compact antibiotic susceptibility test revealed the antibiotic resistance, extended-spectrum β-lactamase (ESBL), and carbapenemase (CP) production phenotypes of the selected Gram-negative bacilli. Clinical and Laboratory Standards Institute (CLSI)-based phenotypic tests for ESBL and CP production revealed discrepant results with the Vitek^®^ 2 Advanced Expert System phenotypes, specifically in some *A. baumannii* and *P. aeruginosa* isolates. Resistance category 1 (RC1) refers to the isolates non-susceptible to third-generation cephalosporins, RC2 refers to those non-susceptible to third-generation cephalosporins and penicillin-β-lactamase inhibitors, RC3 to those resistant to carbapenems and other β-lactams, and RC4 to those with clinically unusual resistance profiles. The antibiotics used in categorizing the isolates are CAZ—ceftazidime, CRO—ceftriaxone, AMC—amoxicillin–clavulanic-acid, TZP—piperacillin–tazobactam, ERT—ertapenem, IPM—imipenem, MEM—meropenem, LVX—levofloxacin. The phenotypes from Vitek^®^ 2 AES are ESBL—extended-spectrum β-lactamase producer, CP—carbapenemase producer. Phenotypes from CLSI-based assays are ESBL producer (tested using disk-diffusion-based ESBL assay), SCP—serine carbapenemase producer (tested using modified carbapenem inactivation method), and MBL—metallo-β-lactamase producer (tested using EDTA-modified carbapenem inactivation methods). “Not tested” means that the antibiotic was not included in the Vitek^®^ 2 antibiogram profiling based on CLSI guidelines.

**Figure 2 pathogens-12-01019-f002:**
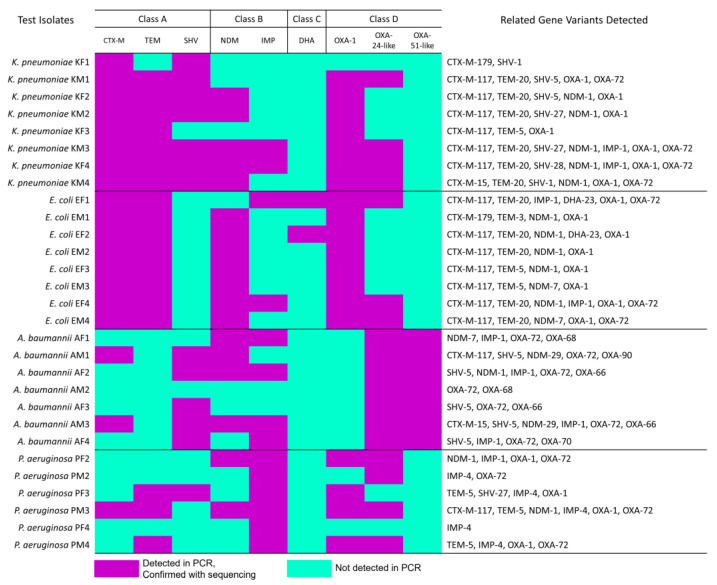
PCR and sequence analysis revealed different classes of β-lactamase genes among the different Gram-negative bacilli tested in the study. Classes described in the figure follow the Ambler classification system. β-lactamase genes presented include: CTX-M—cefotaximase-Munich, TEM—Temoneira β-lactamase, SHV—sulfhydryl reagent variable β-lactamase, NDM—New Delhi metallo-β-lactamase, IMP—imipenemase, DHA—Dhahran β-lactamase, OXA—oxacillinase. Related gene variants were based on maximum likelihood trees generated for each gene sequence (Appendix A).

**Figure 3 pathogens-12-01019-f003:**
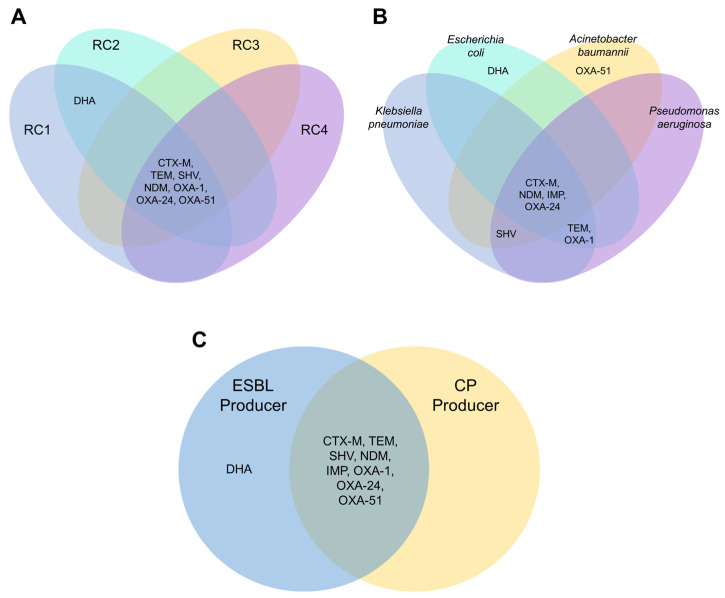
Venn diagrams show the distributional pattern of the β-lactamase genes tested in the study when compared between (**A**) resistance categories, (**B**) bacterial species, and (**C**) β-lactamase produced. Resistance category 1 (RC1) refers to the isolates non-susceptible to third-generation cephalosporins but not to other β-lactams, RC2 refers to those non-susceptible to third-generation cephalosporins and β-lactam–β-lactamase inhibitor combinations, RC3 to those resistant to carbapenems and other β-lactams, and RC4 to those with clinically unusual resistance profiles. Acronyms in panel C: ESBL—extended-spectrum β-lactamase, CP—carbapenemase. β-lactamase genes presented include: CTX-M—cefotaximase-Munich, TEM—Temoneira β-lactamase, SHV—sulfhydryl reagent variable β-lactamase, NDM—New Delhi metallo-β-lactamase, IMP—imipenemase, DHA—Dhahran β-lactamase, OXA—oxacillinase.

## Data Availability

The data presented in this study are available in the manuscript and its Appendix A.

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
