# Peer review of "Clinically Isolated β-Lactam-Resistant Gram-Negative Bacilli in a Philippine Tertiary Care Hospital Harbor Multi-Class β-Lactamase Genes"

_pathogens, 2023, doi:10.3390/pathogens12081019_

Round 1

Reviewer 1 Report

Authors analyzed 29 GNR isolates in Philippines for the presence of beta-lactamase genes. Authors analysing approach by classifying A-D categories seems to be interesting, this manuscript has critical concern.

As epidemiological study, 29 isolates of four species are  too small in number, and epidemiological prevalence of beta-lactamase genes are not able to be determined. Actually there is no novel findings.  

Author Response

Response to Reviewer 1 Comments

Point 1: Authors analyzed 29 GNR isolates in the Philippines for the presence of beta-lactamase genes. Authors analyzing approach by classifying A-D categories seems to be interesting, but this manuscript has critical concern. As an epidemiological study, 29 isolates of four species are too small in number, and the epidemiological prevalence of beta-lactamase genes is not able to be determined. Actually, there are no novel findings.

Response 1: We thank the reviewer for the constructive review of our manuscript. To clarify, we did not intend to determine the prevalence of the tested β-lactamase genes among our isolates as we agree that it would require a larger collection for us to do so. Instead, we wanted to preliminarily investigate the presence of multiple β-lactamase genes among isolates with resistance phenotypes considered clinically critical. This primary goal stemmed from our initial literature review that showed a lack of reporting on the co-occurrence of β-lactamase genes in Philippine Gram-negative isolates. We see this aspect as an unexplored area of research locally even though the National Antibiotic Resistance Surveillance Report showed a high prevalence of critical resistance phenotypes (Antimicrobial Resistance Surveillance Program Annual Report 2022, 2023; https://arsp.com.ph/publications/). Through our investigation, we were able to contribute novel findings such as (1) the first report on the detection of multi-class BLs among β-lactam-resistant Gram-negative bacilli in a Philippine hospital, (2) the discovery of the presence of cryptic metallo-BLs among certain carbapenem-susceptible isolates, and (3) the initial report of an unusual carbapenem-resistant but cephalosporin-susceptible A. baumannii.

Meanwhile, in terms of the sample size, we admit that we limited our analyses to only 29 isolates based on the selection criteria that we set. However, we would like to qualify our sample size by citing other publications that performed similar molecular genotyping investigations. For instance, several recent publications used less than 30 isolates in the genotypic characterization of selected Gram-negative bacteria (Davis et al., 2011; doi: 10.1128/AEM.02588-10; Shawky et al., 2021; doi: 10.22207/JPAM.15.4.49; Tufa et al., 2022; doi: 10.1186/s13756-022-01053-7). These investigations show the value of phenotypic/ genotypic characterizations in their respective context. In our case, given that we aim to preliminarily describe the multi-class β-lactamase genes among the Philippines isolates, we believe that our initial report will still serve as a good foundation for further investigations on β-lactam resistance in the Philippines.

We improved our data presentation and included more discussion and limitation points in our revised manuscript. We hope that the updated manuscript further highlighted the contributions of our research toward a positive assessment by the reviewer. 

Reviewer 2 Report

This is a highly technical paper, providing a wealth of molecular information but little perspective for a clinician or microbiologist/epidemiologist.  Since the authors rightly point out the lack of good molecular epidemiology data derived from the Philippines,  the data presented is too focused and lacks sufficient perspective on beta-lactam antibiotic resistance. The following comments/suggestions should/could be addressed:

1. your "categorization" of beta-lactam antibiotic characteristics is not well referenced. References 21-24 do not provide any consensus definitions that is the premise for your selection of clinical isolates. Even if you do clarify the categories, using groups of A-C can be easily confused with the Ambler Classifications A-C of BLAs.  Make clear the difference.

2. selection of isolated.  You state you selected the first 2 clinical isolates per species for each of the 4 categories.  There could have been 32 but you only present data on 29. presumably because your categories are not equally distributed.  But the point is these 29 isolates do not "represent" AMR in the Philippines and this should be clarified in the discussion.  These are highly selected isolates, 29 out of thousands, and it's not clear what the value is of your categorization.

3. molecular probes: while your methods description is good and the results presentation is also, you are using predetermined PCR sequences.  There are over 3,000 BLAs.  Your discussion of discrepancies between methods does not address the presence of chromosomal BLAs, which are particularly frequent in P. aeruginosa and A. baumannii, especially chromosomal AmpCs e.g. PDC and ADC.  Furthermore, the numbers of copies of chromosomal enzymes may be reflective in phenotypic resistance. Limitations need to be discussed.

4. Resistance mechanisms are not limited to BLAs.  There is no mention of porin entry of efflux mutations which result in carbapenem resistance and may explain discrepancies between resistance phenotype and molecular resistance. Only through WGS can this be described and this limitation should be discussed.

5. Antibiotic phenotype.  The only place the S/I/R is presented is in the supplemental table S4.  This should be brought forward to the main manuscript to better characterize the selected isolates.

Author Response

Response to Reviewer 2 Comments

Point 1: This is a highly technical paper, providing a wealth of molecular information but little perspective for a clinician or microbiologist/epidemiologist.  Since the authors rightly point out the lack of good molecular epidemiology data derived from the Philippines, the data presented is too focused and lacks sufficient perspective on beta-lactam antibiotic resistance. The following comments/suggestions should/could be addressed: 1. your "categorization" of beta-lactam antibiotic characteristics is not well-referenced. References 21-24 do not provide any consensus definitions that are the premise for your selection of clinical isolates. Even if you do clarify the categories, using groups of A-C can be easily confused with the Ambler Classifications A-C of BLAs.  Make clear the difference.

Response 1: We thank the reviewer for the constructive review of our manuscript. To give a better perspective on β-lactam antibiotic resistance and its implications for clinicians and epidemiologists in the Philippine setting, we included additional background information on the study’s context in both the Introduction and Discussions sections. Specifically, we added the following: (1) updated the antibiotic resistance surveillance data to reflect the recent 2022 report published by the Philippine Department of Health, (2) additional information on β-lactam resistance mechanisms, and (3) implications of β-lactam resistance in clinical management and infection control. These additional sections can be found on page 1 lines 41-45, page 2 lines 55-70, and page 8 lines 301-307.

In terms of the resistance categorization, we would like to further qualify the categories by referring to the recent reports from the National Antibiotic Resistance Surveillance Program (ARSP) of the Philippine Department of Health (Antimicrobial Resistance Surveillance Program Annual Report 2022, 2023; https://arsp.com.ph/publications/). Specifically, we presented the recent statistics on the representation of these resistance phenotypes among local isolates. The chosen resistance categories were also vouched by our consulting infectious diseases specialist who helped in the final conceptualization of our study. We agree that there is no consensus on the “critical/ significant” resistance categories in global literature. We believe that this is because of the differences in patient profiles, infection control practices, and pathogen clonalities among various healthcare institutions. However, by referring to the ARSP report for choosing the appropriate resistance categories, we hope to have captured what was generally applicable in the Philippine setting, with the addition of resistance category 4 (RC4) considered as “unusual”. RC4 was included in our study as isolates with this phenotype may have been frequently missed in the clinical laboratory. The additional details for each resistance category can be found on page 3, lines 102-115.

Finally, to avoid confusion between the resistance groups and the Ambler classification, we renamed our categories as “Resistance Categories 1-4”. As indicated in the manuscript, “Resistance Category 1 (RC1)” replaced “Category A”, “RC2” replaced “Category B”, “RC3” replaced “Category C”, and “RC4” replaced “Category D.” The names of the strains were also changed to indicate their respective resistance categories (i.e., K. pneumonaie under RC1 is now named K. pneumoniae KF1, KM1, etc.). These updated categories and strain naming can be found in the entirety of the manuscript and the supplementary file.

Point 2: Selection of isolates. You state you selected the first 2 clinical isolates per species for each of the 4 categories.  There could have been 32 but you only presented data on 29. Presumably, because your categories are not equally distributed.  But the point is these 29 isolates do not "represent" AMR in the Philippines and this should be clarified in the discussion.  These are highly selected isolates, 29 out of thousands, and it's not clear what the value is of your categorization.

Response 2: We thank the reviewer for the technical assessment of the sampling and isolate selection. We confirm that there is an unequal distribution among the collected isolates per resistance category, resulting in the limited number of strains analyzed. We highlighted this aspect as part of our limitation. Meanwhile, we agree that the chosen isolates do not necessarily represent AMR in the Philippines. However, as we have discussed in our response to your previous point, the categories were chosen to capture the significant phenotypes as reported in the National Antibiotic Resistance Surveillance (ARSP) report. Therefore, the analysis of the 29 strains could provide preliminary insights into the potential uninvestigated characteristics of the local pathogen strains. Evidently, from the 29 isolates, we detected multi-class BLs among β-lactam-resistant Gram-negative bacilli in a Philippine hospital, discovered the presence of cryptic metallo-BLs among certain carbapenem-susceptible isolates, and reported a case of an unusual carbapenem-resistant but cephalosporin-susceptible A. baumannii. All these findings were not reported in any previous local investigations, opening opportunities for epidemiologists and microbiologists to investigate these strains further. We added this additional context as part of our discussion, which can be found on pages 8-9, lines 313-323 of the revised manuscript.

Point 3: Molecular probes: while your methods description is good and the presentation of the results is also, you are using predetermined PCR sequences.  There are over 3,000 BLAs.  Your discussion of discrepancies between methods does not address the presence of chromosomal BLAs, which are particularly frequent in P. aeruginosa and A. baumannii, especially chromosomal AmpCs e.g. PDC and ADC. Furthermore, the number of copies of chromosomal enzymes may be reflective in phenotypic resistance. Limitations need to be discussed.

Response 3: We recognize this important comment from the reviewer. As with any other PCR-based assays, we are only limited to the chosen target genes. We highlighted this aspect in our limitations. We chose to focus on specific transferable/ acquired β-lactamase genes as these were found to have significant implications in the spread of resistance in the hospital setting (De Angelis et al., 2020; doi: 10.3390/ijms21145090). We agree that chromosomal β-lactamases (particularly AmpCs) can play roles in the observed phenotype of P. aeruginosa and A. baumannii. Hence, we added this aspect as part of our discussion (explaining the limits of the chosen phenotypic assay), and the analysis of chromosomal β-lactamase genes as part of our limitations. These additional statements can be found on page 2 lines 68-70, page 4 lines 157-158, page 9 lines 337-340, and page 10 lines 410-413 of the revised manuscript.

Point 4: Resistance mechanisms are not limited to BLAs.  There is no mention of porin entry of efflux mutations which result in carbapenem resistance and may explain discrepancies between resistance phenotype and molecular resistance. Only through WGS can this be described, and this limitation should be discussed.

Response 4: We thank the reviewer for reiterating the need to discuss the other mechanisms of resistance in explaining the observed phenotypes of the isolates. We added this aspect in our updated discussion. Specifically, we reiterated that other mechanisms of carbapenem resistance not measured by the chosen phenotypic test, including high levels of chromosomal BLs expression, porin loss, or efflux pumps, may have contributed to the discrepancy between the assay results and the isolate’s resistance profiles as supported in the literature (Lee et al., 2017; doi: 10.3389/fcimb.2017.00055). This insight also highlights the unreliability of current phenotypic tests in assessing β-lactamase production in A. baumannii and P. aeruginosa, hence the lack of recommended tests in both EUCAST and CLSI guidelines.

Meanwhile, we appreciate the reviewer’s reiteration of the value of whole genome sequencing in further understanding the complexity of resistance mechanism determination in highly resistant pathogens. Although we already initially included this aspect in our limitation and recommendation, we further highlighted them in the revised manuscript. The manuscript updates relating to this point can be found on page 9 lines 337-340, and page 10 lines 410-413.

Point 5: Antibiotic phenotype.  The only place the S/I/R is presented is in the supplemental table S4.  This should be brought forward to the main manuscript to better characterize the selected isolates.

Response 5: We thank the reviewer for this technical suggestion. However, we would like to clarify that the antibiotic resistance phenotypes were also presented in Figure 1 as a heatmap. To further qualify the S/I/R phenotypes, we explicitly annotated these codes within the heatmap. The revised heatmap now clearly shows the resistance patterns of each of the isolates against the relevant antibiotics under their respective resistance categories. Supplementary Table S4 still remained in our supplementary file in case the readers want to view the complete antibiogram profiles of the isolates. The revised Figure 1 can be found on page 5, line 205. 

Reviewer 3 Report

Moderate comments:

Lines 24-27. The rate cannot be represented as a percentage of such a few strains. It is better to present in the format (n/29). This applies to the entire text.

Line 54. (CP) should be replaced on (CPs). BLs must be explained at the first use.

Lines 58-60. This citation is not accurate from the Reference 9. Please specify.

Lines 69-70. “differentiate the different” is not a beautiful phrase.

Figure 1. It is not clear the attribution of some isolates to the manifested categories: A. baumannii A1, B1, B2, D1.

Figure 2. Non-specific amplification should be removed from the picture. “OXA-24” and “OXA-51” should be replaced on “OXA-24-like” and “OXA-51-like”.

Line 206 and other points: beta-lactamase genes should be presented in format blaTEM

References are not presented under the Journal’s requirements.

Author Contributions are not in accordance to the Journal’s requirements.

Author Response

Response to Reviewer 3 Comments:

Point 1: Lines 24-27. The rate cannot be represented as a percentage of such a few strains. It is better to present in the format (n/29). This applies to the entire text.

Response 1: We thank the reviewer for this important suggestion. We updated the data presentation to follow the suggested format (n/29) as applicable in the text. The updates can be found in the entirety of the revised manuscript.

Point 2: Line 54. (CP) should be replaced on (CPs). BLs must be explained at the first use.

Response 2: We thank the reviewer for pointing out this abbreviation usage. We already replaced CP with CPs on the indicated position and added the context for the first time usage of the abbreviation “BLs”. These updates can be found on page 2 lines 54 and 58.

Point 3: Lines 58-60. This citation is not accurate from Reference 9. Please specify.

Response 3: We thank the reviewer for this technical comment. The original citation (Sawa et al., 2020; doi: 10.1186/s40560-020-0429-6) is a review article that discusses the Ambler classification system. We revised the citation to now refer to the original paper that developed the Ambler classification system (Ambler, 1980; doi:10.1098/rstb.1980.0049). The new citation can be found on page 2 line 66, and the new reference can be found on page 12 line 513.

Point 4: Lines 69-70. “differentiate the different” is not a beautiful phrase.

Response 4: We thank the reviewer for the comment on the manuscript wording. We agree with the comment, and we reworded the statement to make it more pleasing to the readers. The sentence now reads “However, these assays have inherent limitations in their diagnostic sensitivity and do not distinguish the different BL classes.” This change can be found on page 2 lines 83-84.

Point 5: Figure 1. It is not clear the attribution of some isolates to the manifested categories: A. baumannii A1, B1, B2, D1.

Response 5: We thank the reviewer for the clarificatory comment. To make the attribution of the isolates to their resistance categories easier, and to prevent the confusion of the Ambler class types with the set resistance categories, we renamed our categories as “Resistance Categories 1-4”. As indicated in the manuscript, “Resistance Category 1 (RC1)” replaced “Category A”, “RC2” replaced “Category B”, “RC3” replaced “Category C”, and “RC4” replaced “Category D.” The names of the strains were also changed to indicate their respective resistance categories (i.e., K. pneumonaie under RC1 is now named K. pneumoniae KF1, KM1, etc.). We updated Figures 1, 2, and 3 to reflect these changes. The revised figures can be found on page 5 line 205, page 7 line 272, and page 8 line 289. The strain names of the isolates were also changed for the rest of the manuscript and the supplementary file.

Point 6: Figure 2. Non-specific amplification should be removed from the picture. “OXA-24” and “OXA-51” should be replaced by “OXA-24-like” and “OXA-51-like”.

Response 6: We thank the reviewer for this technical suggestion. We removed the non-specific amplifications from the figure and replaced the “OXA-24” and “OXA-51” headers with “OXA-24-like” and “OXA-51-like.” The revised figure can be found on page 7 line 272.

Point 7: Line 206 and other points: beta-lactamase genes should be presented in format blaTEM

Response 7: We thank the reviewer for pointing out this technical error. We already updated the naming of the genes to follow the correct format. The updates can be found in the entirety of the revised manuscript.

Point 8: References are not presented under the Journal’s requirements.

Response 8: We thank the reviewer for pointing out this formatting error. We already updated our reference list following the prescribed format of the journal. The updated reference list can be found on pages 12-14 lines 490-598.

Point 9: Author Contributions are not in accordance with the Journal’s requirements.

Response 9: We thank the reviewer for pointing out this formatting error. We already updated the “Author Contributions” section of our manuscript to comply with the requirements of the journal. The update can be found on page 11 lines 464-469.

Round 2

Reviewer 1 Report

I understood the authors' aim in this study.